# A Brief Overview of AI Governance for Responsible Machine Learning Systems

**Navdeep Gill**     **Abhishek Mathur**     **Marcos V. Conde**[*]

**H2O.ai**
Mountain View, CA
{navdeep.gill, abhishek.mathur, marcos.conde}@h2o.ai

## Abstract

Organizations of all sizes, across all industries and domains are leveraging artificial intelligence (AI) technologies to solve some of their biggest challenges around operations, customer experience, and much more. However, due to the probabilistic nature of AI, the risks associated with it are far greater than traditional technologies. Research has shown that these risks can range anywhere from regulatory, compliance, reputational, and user trust, to financial and even societal risks. Depending on the nature and size of the organization, AI technologies can pose a significant risk, if not used in a responsible way. This position paper seeks to present a brief introduction to AI governance, which is a framework designed to oversee the responsible use of AI with the goal of preventing and mitigating risks. Having such a framework will not only manage risks but also gain maximum value out of AI projects and develop consistency for organization-wide adoption of AI.

## 1 Introduction

In this position paper, we share our insights about AI Governance in companies, which enables new connections between various aspects and properties of trustworthy and socially responsible Machine Learning: security, robustness, privacy, fairness, ethics, interpretability, transparency, etc.

For a long time *Artificial intelligence* (AI) was something enterprise organizations adopted due to the huge amounts of resources they have at their fingertips. Today, smaller companies are able to take advantage of AI due to newer technologies, *e.g.* cloud software, which are significantly more affordable than what was available in the past [3, 13, 17, 19]. AI has been on an upward trajectory in recent years and it will increase significantly over the next several years [30] [22] [39]. However, every investment has its pros and cons. Unfortunately, the cons associated with AI adoption are caused by its inherent uncertainties, and the builders of such AI systems who do not take the necessary steps to avoid problems down the road [28, 19]. Note that, in this work, AI comprises modern *Machine Learning* (ML) and *Deep Learning* (DL) systems, yet not their software around them - which represents other threats and vulnerabilities by itself [29, 1, 16] -.

### 1.1 Problems within Industry

Some of the most popular applications of AI are: anomaly detection and forecasting [27, 44], recommender systems [38], medical diagnosis [45, 18], earth science [11, 10], and search engines[12].

However, these applications of AI in industry is still in its infancy. With that said, many problems have arisen since its adoption [15, 19, 28], which can be attributed to several factors:

---

[*]MC is also with University of Würzburg, CAIDAS. Supported by The Alexander von Humboldt Foundation.

36th Conference on Neural Information Processing Systems (NeurIPS 2022).

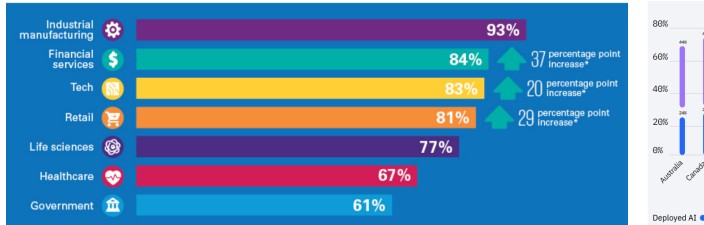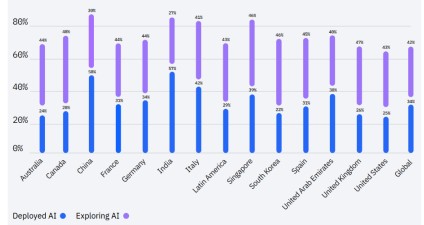

Figure 1: (Left) "Rate of AI adoption skyrocketed during COVID-19" by KPMG [23]. (Right) IBM Global AI Adoption Index 2022 [22]. We refer the reader to the surveys [23, 22] for more details.

**Lack of risk awareness and management:** Too much attention is given to applications of AI and its potential success and not enough attention is given to its potential pitfalls and risks.

**AI adoption is moving too fast:** According to a survey by KPMG in 2021 [23], many respondents noted that AI technology is moving too fast for their comfort in industrial manufacturing (55%), technology (49%), financial services (37%), government (37%), and health care (35%) sectors.

**AI adoption needs government intervention:** According to the same survey by KPMG, [23] (see Figure 1), an overwhelming percentage of respondents agreed that governments should be involved in regulating AI technology in the industrial manufacturing (94%), retail (87%), financial services (86%), life sciences (86%), technology (86%), health care (84%), and government (82%) sectors.

**Companies are still immature when it comes to adopting AI:** Some companies are not prepared for business conditions to change once a ML model is deployed into the real world.

Many of these problems can be avoided with proper governance mechanisms. AI without such mechanisms is a dangerous game with detrimental outcomes due its inherent uncertainty [49, 8]. With that said, adding governance into applications of AI is imperative to ensure safety in production.

## 2   AI Governance

**What is AI Governance (AIG)?**

> AI Governance is a framework to operationalize responsible artificial intelligence at organizations. This framework encourages organizations to curate and use bias-free data, consider societal and end-user impact, and produce unbiased models; the framework also enforces controls on model progression through deployment stages. The potential risks associated with AI need to be considered when designing models, before they affect the quality of models and algorithms. If left unmonitored, AI may not only produce undesirable results, but can also have a significant adverse impact on the organization.

In order for organizations to realize the maximum value out of AI projects and develop consistency for organization-wide adoption of AI, while managing significant risks to their business, they must implement AI governance [41, 14, 20, 24]; this enables organizations to not only develop AI projects in a responsible way, but also ensure that there is consistency across the entire organization and the business objective is front and center. With the AI governance implemented (as illustrated in Figure 2), the following benefits can be realized:

**Alignment and Clarity:** teams would be aware and aligned on what the industry, international, regional, local, and organizational policies are that need to be adhered to.

**Thoughtfulness and Accountability:** teams would put deliberate effort into justifying the business case for AI projects, and put conscious effort into thinking about end-user experience, adversarial impacts, public safety & privacy. This also places greater accountability on the teams developing their respective AI projects.

**Consistency and Organizational Adoption:** teams would have a more consistent way of developing and collaborating on their AI projects, leading to increased tracking and transparency for their projects.

This also provides an overarching view of all AI projects going on within the organization, leading to increased visibility and overall adoption.

**Process, Communication, and Tools:** teams would have complete understanding of what the steps are in order to move the AI project to production to start realizing business value. They would also be able to leverage tools that take them through the defined process, while being able to communicate with the right stakeholders through the tool.

**Trust and Public Perception:** as teams build out their AI projects more thoughtfully, this will inherently build trust amongst customers and end users, and therefore a positive public perception.

AI governance requires the following:

1. A structured organization that gives AIG leaders the correct information they need to establish policies and accountability for AI efforts across their entire organization. For smaller organizations, this might require a more phased approach in which they will work towards the desired structural framework of AIG. For larger organizations, this process might be more attainable due to resources alone, *e.g.* people, IT infrastructure, larger budgets.

2. A concrete and specific AI workflow that collects information needed by AIG leaders will help enforce the constructed policies. This provides information to various parties in a consumable manner. Having such information can be used to minimize mistakes, errors, and bias, amongst other things.

The requirements for AI governance manifest into a framework that an organization must work towards developing. The components of this framework need to be transparent and comprehensive to achieve a successful implementation of AIG. Specifically, this should focus on organizational and use case planning, AI development, and AI "operationalization", which come together to make a **4 stage AI life cycle approach**.

## 3 Stages of a Governed AI Life cycle

### 3.1 Organizational Planning

An AI Governance Program [24, 14, 48] should be organized in such a way that (a) there is comprehensive understanding of regulations, laws, and policies amongst all team members (b) resources and help available for team members who encounter challenges (c) there is a light weight, yet clear process to assist team members.

1. **Regulations, Laws, Policies**

   Laws and regulations that apply to a specific entity should be identified, documented, and available for others to review and audit. These regulations, laws, and policies vary across industry and sometimes by geographical location. Organizations should, if applicable, develop policies for themselves, which reflect their values and ethical views [48, 33, 19]; this enables teams to be more autonomous and make decisions with confidence.

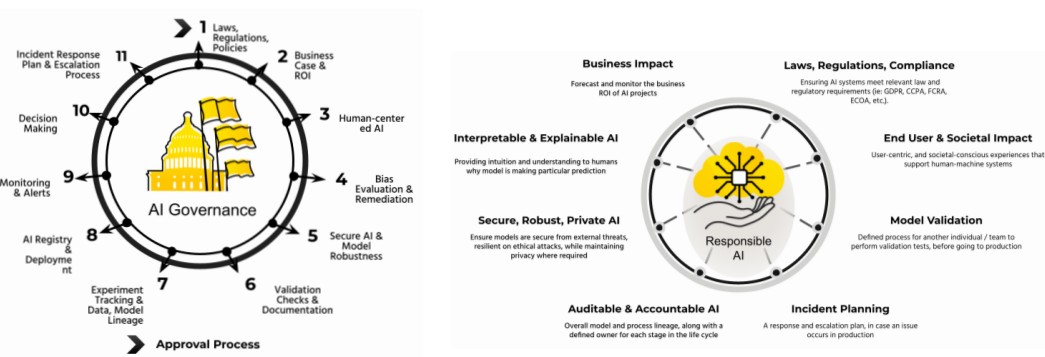

Figure 2: Illustration of the AI Governance application towards responsible AI in companies.

2. **Organization (Center of Competency)**

   Establishing groups within an organization that provide support to teams with AI projects can prove to be quite beneficial. This includes a group that is knowledgeable with regulations, laws, and policies and can answer any questions that AI teams may have; a group that is able to share best practices across different AI teams within the organization; a group that is able to audit the data, model, process, etc. to ensure there isn't a breach or non-compliance. For more information, we refer the reader to to the survey by Floridi *et al.* [19].

3. **Process**

   Developing a light-weight process that provides guidelines to AI teams can help with their efficiency, rather than hinder their progress and velocity. This involves identifying what the approval process and incident response would be for data, model, deployments, etc.

## 3.2 Use Case Planning

Building use cases involves establishing business value, technology stack, and model usage. The group of people involved in this process can include: subject matter experts, data scientists/analysts/annotators and ML engineers, IT professionals, and finance departments.

**Business Value Framework.** The AI team should ensure that the motivation for the AI use case is documented and communicated amongst all stakeholders. This should also include the original hypothesis, and the metrics that would be used for evaluating the experiments.

**Tools, Technology, Products.** The AI team should either select from a set of pre-approved tools and products from the organization or get a set of tools and products approved before using in an AI user case. If tools for AI development are not governed, it not only leads to high costs and inability to manage the tools (as existing IT teams are aware), it also leads to not being able to create repeatability and traceability into AI models.

**Model Usage.** Once a sense of value is attached to the use case, then the next step would be to break down the use case to its sub-components which include, but are not limited to, identifying the consumer of the model, the model's limitations, and potential bias that may exist within the model, along with its implications. Also, one would want to ensure inclusiveness of the target, public safety/user privacy, and identification of the model interface needed for their intended use case.

## 3.3 AI Development

Development of a machine learning model, including data handling and analysis, modeling, generating explanations, bias detection, accuracy and efficacy analysis, security and robustness checks, model lineage, validation, and documentation.

1. **Data Handling, Analysis and Modeling**

   The first technical step to any AI project is the procurement and analysis of data, which is critical as it lays the foundation for all work going forward. Once data is analyzed, then one must decipher if modeling is needed for the use case at hand. If modeling is needed, then the application of AI can take place. Such an application is an iterative process spanned across many different types of people.

2. **Explanations and Bias**

   The goal of model explanations is to relate feature values to model predictions in a human-friendly manner [34]. What one does with these explanations breaks down to 3 personas: modeler, intermediary user, and the end user. The modeler would use explanations for model debugging and gaining understanding of the model they just built. The intermediary user would use what the modeler made for actionable insights. And finally, the end user is the person the model affects directly. For these reasons, Explainable Artificial Intelligence (**XAI**) is a very active research topic [5, 21].

   Bias, whether intentional (disparate treatment) or unintentional (disparate impact), is a cause for concern in many applications of AI [42, 31]. Common things to investigate when it comes to preventing bias include the data source used for the modeling process, performance issues amongst different demographics, disparate impact, identifying known limitations &

potential adverse implications, and the models impact on public safety [6]. We refer the reader to the survey by Mehrabi *et al.* [31] for more details about bias and fairness in AI.

3. **Accuracy, Efficacy, & Robustness**

   Accuracy of a machine learning model is critical for any business application in which predictions drive potential actions. However, it is not the most important metric to optimize. One must also consider the efficacy of a model, *i.e.* is the model making the intended business impact?

   When a model is serving predictions in a production setting, the data can be a little or significantly different from the data that the project team had access to. Although model drift and feature drift can capture this discrepancy, it is a lagging indicator, and by that time, the model has already made predictions. This is where **Robustness** comes in: project teams can proactively test for model robustness, using "out of scope" data, to understand whether the model perturbs. The out of scope data can be a combination of manual generation (toggle with feature values) and automatic generation (system toggles feature values).

4. **Security**

   ML systems today are subject to general attacks that can affect any public facing IT system [7], [36]; specialized attacks that exploit insider access to data and ML code; external access to ML prediction APIs and endpoints [46], [43]; and trojans that can hide in third-party ML artifacts. Such attacks must be accounted for and tested against before sending a machine learning model out into the real world.

5. **Documentation & Validation**

   An overall lineage of the entire AI project life-cycle should be documented to ensure transparency and understanding [32], [40], which will be useful for the AI team working on the project and also future teams who must reference this project for their own application.

   Model validation [35, 25] is the set of processes and activities that are carried out by a third party, with the intent to verify that models are robust and performing as expected, in line with the business use case. It also identifies the impact of potential limitations and assumptions. From a technical standpoint, the following should be considered: (i) Sensitivity Analysis. (ii) In-sample vs. Out-of-sample performance. (iii) Replication of results from model development team. (iv) Stability analysis. Model "validators" should document all of their findings and share with relevant stakeholders.

## 3.4 AI Operationalization

Deploying a machine learning model into production (*i.e.* MLOps [2, 47]) is the first step to potentially receiving value out of it. The steps that go into the deployment process should include the following:

**Review-Approval Flow:** Model building in an AI project will go through various stages: experimentation, model registration, deployment, and decommissioning. Moving from one stage to the next would require "external" reviewer(s) who will vet and provide feedback.

**Monitoring & Alerts:** Once a model is deployed, it must be monitored for various metrics to ensure there is not any degradation in the model. The cause for a model degrading when deployed can include the following: feature and/or target drift, lack of data integrity, and outliers, amongst other things. In terms of monitoring, accuracy, fairness, and explanations of predictions are of interest [4, 26].

**Decision Making:** The output of a machine learning model is a prediction, but that output must be turned into a decision. How to decide? Will it be autonomous? Will it involve a human in the loop? The answers to these questions vary across different applications, but the idea remains the same, ensuring decisions are made in the proper way to decrease risk for everyone involved.

**Incident Response and Escalation Process:** With AI models being used in production, there is always going to be a chance for issues to arise. Organizations should have an incident response plan and escalation process documented and known to all project teams.

Companies who successfully implement AI governance for AI applications will result in a highly impactful use of artificial intelligence. While those who fail to do so, risk catastrophic outcomes and an arduous road to recovery as shown in the following use-case.

# 4 AIG Use Case

In this section we describe a recent use case where - we believe - AI Governance could have avoided a terrible outcome.

In 2021, the online real estate technology giant, Zillow, shut down its **AI-powered** house-flipping business. At it's core, this line of business relied heavily on forecasting from their machine learning models. Zillow found itself overpaying for homes due to overestimating the price of a property. Such overestimation of property values led to a loss of $569 million, and 28% of their valuation[9, 37]. The monetary loss and laying off of 2,000 employees also came with a reputational cost.

This **AI failure** begs the question "What mistakes did Zillow make?" and "Could this have been prevented?". The answer to the second question is a firm "yes". However, the answer to the first question has many facets, but we try to break down the key mistakes below:

1. **Removing "Human in the Loop"**

   In its early days, this company hired local real estate agents and property experts to verify the output of their home price prediction ML algorithm; however, as the business scaled, the human verification process around the algorithms was minimized[9] and the offer-making process was automated, which helped to cut expenses and increase acquisitions. However, removing human verification in such a volatile domain led to predictions taken at face value without any verification, which played a big role in overpricing of properties.

   Keeping the human in the loop can help companies from relying on overestimated and biased predictions. Considering a ML ecosystem for decision-making in organizations, having a human in the loop is a safety harness and should be used in any high stake decision making.

2. **Not Accounting for Concept Drift and Lack of Model Monitoring**

   Taking a look back at the timeline of events, it appears that the ML algorithms were not adjusted accordingly to the real market status[9]. The algorithms continued to assume that the market was still "hot" and overestimated home prices.

   To avoid problems of drift, specifically concept drift, companies should leverage tools for monitoring and maintaining the quality of AI models. Ideally, in this example, they should have set up an infrastructure that automatically alerts data science teams when there is drift or performance degradation, support root cause analysis, and inform model updates with humans-in-the-loop.

3. **Lack of Model Validation**

   Before organizations deploy any of their algorithms, they should have enact a set of processes and activities intended to verify that their models are performing as expected and that they are in line with the business use case. Effective validation ensures that models are robust. It also identifies potential limitations and assumptions, and it assesses their possible impact. A lack of model validation played a crucial role in this case, and led to the overestimation of home prices. It should be noted that -ideally- model validation needs to be carried out by a third party that did not take part in the model building process.

4. **Lack of Incident Response**

   It is not clear when this company started to realize that their model's were degrading and producing erroneous results. The lack of having such an incident response plan can cost businesses an exuberant amount of money, time, and human resources. In retrospect, if proper AI Governance would had been implemented, the company could have started an incident response *i.e.* incident identification and documentation, human review, and redirection of model traffic. These three steps alone could have prevented a lot of damage to their business.

This industry use case is an example of what can happen without proper AI governance. The mistakes that were made could have been avoided if proper AI governance principles were taken into account from the inception of their use case. Specifically, if this company continued to rely on their subject matter experts, accounted for various types of drift, added model validation, and implemented concrete model monitoring with proper incident response planning, then this whole situation could have been avoided or the damage could have been at a much lower magnitude.

# 5 Conclusion

AI systems are used today to make life-altering decisions about employment, bail, parole, and lending, and the scope of decisions delegated by AI systems seems likely to expand in the future. The pervasiveness of AI across many fields is something that will not slowdown anytime soon and organizations will want to keep up with such applications. However, they must be cognisant of the risks that come with AI and have guidelines around how they approach applications of AI to avoid such risks. By establishing a framework for AI Governance, organizations will be able to harness AI for their use cases while at the same time avoiding risks and having plans in place for risk mitigation, which is paramount.

**Social Impact**   As we discuss in this paper, governance and certain control over AI applications in organizations should be mandatory. *AI Governance* aims to enable and facilitate connections between various aspects of trustworthy and socially responsible machine learning systems, and therefore it accounts for security, robustness, privacy, fairness, ethics, and transparency. We believe the implementation of these ideas should have a positive impact in the society.

**Acknowledgements**   We thank the Trustworthy and Socially Responsible Machine Learning (TSRML) Workshop at NeurIPS 2022. This work was supported by H2O.ai.

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
