# OpenReview forum: "A Brief Overview of AI Governance for Responsible Machine Learning Systems"
_NeurIPS.cc/2022/Workshop/TSRML — TSRML2022_

### Official Review · Reviewer_ALZt · 2022-10-20
**Decent overview although lacks major significance**

**Overall Rating:** 6

**Summary:**

The paper provides an overview to the topic of AI governance, a framework designed to oversee the responsible use of AI aiming to prevent and mitigate risks. The work talks about the current problems in the use of AI adoption in the industry and how organizations can help manage risks by adopting AI governance.

**Strengths:**

- The work tackles an important topic of AI governance which is definitely needed in the industrial uses of AI going forward.
- The paper well illustrates the current problems and risks within the industry and how AI governance can help overcome them.
- The paper is well written, comprehensive and adequately explains the concept of AI governance to someone not familiar with it.

**Weaknesses:**

The major weakness is that the works lacks novelty and major significance. While the work gives a high level overview of why the industry should adopt governance and enlists the major benefits of it, there is little value addition of this work in current form. I would had liked to see the authors tackle some specific company or a specific problem and detailing the exact procedure from step A to Z which it should take to successfully implement AI governance to solve that problem and how is it better than what it was to begin with.

**Overall Recommendation:**

Overall, the paper does a decent job of introducing the concept of AI governance. Although, I am a bit skeptical of the significance of the work I'm okay with it being presented in the workshop since I feel AI governance is an important topic which needs to be discussed more often going forward.

**Review Confidence:**

2: The reviewer is willing to defend the evaluation, but it is quite likely that the reviewer did not understand central parts of the paper

---

### Official Review · Reviewer_FBHJ · 2022-10-20

**Overall Recommendation:** See above.
**Overall Rating:** 6

**Summary:**

This position paper aims to introduce AI governance such that it can be used in a responsible way.

**Strengths:**

+ The research problem studied in this paper is important.
+ How to perform AI governance is still an open challenge.

**Weaknesses:**

- Lack of details. The paper only lists some high-level directions. It would be good if some concrete strategies could be discussed.

**Review Confidence:**

3: The reviewer is fairly confident that the evaluation is correct

---

### Official Review · Reviewer_dpsD · 2022-10-21
**Interesting guide, but not sure about the significance**

**Overall Rating:** 5

**Summary:**

The paper aims to serve an overview of the AI Governance for Responsible ML systems.
The paper does set some structure to the issue of AI Governance in the industry.
Then it highlights the planning, development, and operation.

While the paper may serve interesting read to those who are very new to the field, the paper seems to provide only very vague overview of the large intricate problem.

**Strengths:**

+ Interesting read for those who may be very new to the problem.

**Weaknesses:**

- Seems to be full of rather vague terminologies and rather humble amount of evidence for what it highlights.

**Overall Recommendation:**

While the paper may serve interesting read to those who are very new to the field, the paper seems to provide only very vague overview of the large intricate problem.

**Review Confidence:**

3: The reviewer is fairly confident that the evaluation is correct

---

### Official Review · Reviewer_7Vx4 · 2022-10-22
**Review for "A Brief Overview of AI Governance for Responsible Machine Learning Systems"**

**Overall Rating:** 7

**Summary:**

The paper discusses and presents a brief overview of governance structures that need to be adopted by companies in order to ensure resposible use and development of AI systems/applications. The paper motivates the need for these governance structures and the benefits it offers to both the companies and the society at large. It then breaks down AI governance into 4 buckets, namely, Organizational Planning, Use Case Planning, AI Development and AI Operationalization and details the sturctures and procedures needed for each bucket. The paper also presents a comprehensive catalog of references and places itself in the context of the rest of the literature.

**Strengths:**

- The paper presents a comprehensive, albeit brief overview of the AI governancy structures that could be adopted within companies. I imagine it would be especially useful for companies that are newly experimenting with using AI in their products/workflows/services.
- The paper also puts the rest of the literature in context allowing the reader to check the specific references if they wish to delve deeper into the topic
- very important work given the increasing adoption of AI by new companies. The document can serve as a useful reference for a lot of those teams being newly established.

**Weaknesses:**

- The paper discusses internal structures and mechanisms that individual companies could adopt. However, most of these structures only work when the responsible deployment are aligned with the company's incentive structures. I would have appreciated discussion of structures that we need at a cross company/institutional level to ensure responsible behavior even when the incentive structures of a specific company itself aren't aligned.
- Likewise, sometimes, even when individual companies might have noble intentions, the market conditions/regulatory environment might not be condusive to responsible behavior. Thus, it is important to also discuss structures and standards that companies would need to collaborate on to modify the market/regulatory environment to encourage responsible behavior.
- A large portion of the AI community is highly pro open-source. I think any company working on AI needs to have an explicit open-sourcing strategy. One could encompass it in the 'values and ethical views', but I think it has become important enough in the community that it deserves a separate discussion. I think it's especially important in the context of deployment given that most existing open source is mostly in the realm of papers and their code. Companies open sourcing (parts of) their AI infrastructure/tools could be a huge contribution for the community.
- A lot of current startups/teams working on AI related applications might eventually die. They also need a strategy to opensource or sell the infrastructure they develop internally to make sure a lot of the effort put into it doesn't go waste. Moreover, the data/insights they might have collected over the course of their project might be sensitive/useful and thus might require a careful handling. Hence, I think 'wrapping up' projects should be explicitly a part of any AI governence strategy.
- I would appreciate if the paper title and abstract better reflect the fact that the paper mainly discusses the AI governance structures that are needed within companies to promote responsible development of AI. My initial impression from reading title and abstract was that the paper would cover a broader set of AI governance questions beyond just the internal structures within a company.

**Overall Recommendation:**

I think the paper provides a useful overview of the AI governance structures and would be useful for new teams and companies looking to venture into using machine learning systems. Thus, I recommend accepting the paper.

**Review Confidence:**

2: The reviewer is willing to defend the evaluation, but it is quite likely that the reviewer did not understand central parts of the paper

---

### Decision · Program_Chairs · 2022-10-23

**Decision:**

Accept

**Comment:**

The paper presents a useful overview of the AI governance and should attract relevant audience in this workshop.